# Design, Synthesis, and Antiproliferative Activity of Selective Histone Deacetylases 6 Inhibitors Containing a Tetrahydropyridopyrimidine Scaffold

**DOI:** 10.3390/molecules28217323

**Published:** 2023-10-29

**Authors:** Bin Wang, Youcai Liu, Lejing Zhang, Yajuan Wang, Zhaoxi Li, Xin Chen

**Affiliations:** 1Department of Biochemistry and Molecular Biology, Sanquan College of Xinxiang Medical University, Xinxiang 453003, China; wangbin@sqmc.edu.cn (B.W.); lejing.2008.cool@163.com (L.Z.); jjxcz@126.com (Y.W.);; 2Experimental Teaching Center of Biology & Basic Medicine, Sanquan College of Xinxiang Medical University, Xinxiang 453003, China; 09513063@sqmc.edu.cn; 3Shaanxi Key Laboratory of Natural Products & Chemical Biology, College of Chemistry & Pharmacy, Northwest A&F University, Xianyang 712100, China

**Keywords:** HDAC6 inhibitor, tetrahydropyridopyrimidine, antitumor, synthesis, selectivity

## Abstract

The development of selective histone deacetylase 6 inhibitors (sHDAC6is) is being recognized as a therapeutic approach for cancers. In this paper, we designed a series of novel tetrahydropyridopyrimidine derivatives as sHDAC6 inhibitors. The most potent compound, 8-(2, 4-bis(3-methoxyphenyl)-5, 8-dihydropyrido [3, 4-*d*]pyrimidin-7(6*H*)-yl)-*N*-hydroxy-8-oxooctanamide (**8f**), inhibited HDAC6 with IC_50_ of 6.4 nM, and showed > 48-fold selectivity over other subtypes. In Western blot assay, **8f** elevated the levels of acetylated *α*-tubulin in a dose-dependent manner. In vitro, **8f** inhibited RPMI-8226, HL60, and HCT116 tumor cells with IC_50_ of 2.8, 3.20, and 3.25 μM, respectively. Moreover, **8f** showed good antiproliferative activity against a panel of tumor cells.

## 1. Introduction

Histone deacetylases (HDACs) are involved in a wide range of biological responses by histone deacetylation and nonhistone lysine post-translational modification and have been identified as targets in the treatment of various diseases, especially for cancer [1,2,3,4,5]. The HDACs family include class I (HDAC1, HDAC2, HDAC3, HDAC8), class II (HDAC4, HDAC5, HDAC6, HDAC7, HDAC9, HDAC10), class III (Sirt1−7), and class IV (HDAC11) [6]. To date, five HDACis have been approved to treat cutaneous T-cell lymphoma, peripheral T-cell lymphoma, or multiple myeloma [7,8,9,10,11]. However, all of them are nonselective or partially selective, which might have potentially toxic side effects [12,13]. In contrast with the lethal effect of HDAC1-3 genetic ablation, mice with HDAC6 knocked out are viable and develop normally [14,15]. HDAC6, mainly located in the cytoplasm, exhibits unique characteristics [16,17]. It contains two tandem catalytic domains and directly acts on a host of cytosolic proteins and substrates such as *α*- and *β*-tubulin, heat shock protein, assembled micro-tubules, and cortactin [18,19,20], which are closely related to tumorigenesis. Moreover, the binding to ubiquitin by the distinctive zinc finger domain makes HDAC6 regulate protein clearance and degradation [21]. The advantage of lower toxicity and improved safety profile has made the development of HDAC6i a hot research topic in cancer treatment [16,22,23,24].

To date, a lot of synthetic sHDAC6is have been reported [25,26,27,28,29,30,31]. The structure of HDAC6i typically contains three parts: (a) a zinc-binding group (ZBG) coordinating with Zn^2+^ ion at the bottom of the active site, (b) a linker region embedding in the hydrophobic tunnel between the catalytic site and the outer surface, and (c) a capping group overlaying on the surface (Figure 1). The clinical ACY-1215 (**1**) inhibited HDAC1 and HDAC6 with IC_50_s of 5 nM and 58 nM and was evaluated for the treatment of multiple myeloma (MM) and lymphoid malignancies [32]. ACY-1215 showed synergistic anti-MM activity together with bortezomib, resulting in protracted endoplasmic reticulum stress and apoptosis. ACY-241 (**2**), similar to the structure of ACY-1215, achieved higher serum concentrations than ACY-1215. The IC_50_ value of ACY-241 against HDAC6 was 2.6 nM, 13~18-fold better than HDAC1-3 [33]. KA2507 (**3**) potently inhibited HDAC6 with IC_50_ of 2.5 nM. It demonstrated antitumor efficacy and immune modulatory effects in preclinical models. In a phase I study, KA2507 showed selective target engagement, no significant toxicities, and prolonged disease stabilization in a subset of patients [34]. Despite great success in sHDAC6is discovery, available clinical agents are still rare, and the lack of therapeutic effect on solid tumors is another problem for HDAC inhibitors.

Because the cap region of the HDAC6 pocket is wider and larger than that of HDAC1 [35], a more rigid and bigger capping group might improve the selectivity toward HDAC6. For HDAC6is **1**–**3**, the common feature is apparent: a “Y” shaped and predominantly aromatic capping group with hydroxamic acid as ZBG. The 5, 6, 7, 8-tetrahydropyrido[3, 4-*d*]pyrimidine (**4**) was frequently used in the development of kinase inhibitors for cancer treatment [36,37]. Hence, the introduction of such a scaffold in one molecule might be beneficial for the anticancer efficacy of HDAC6is. In this paper, we replaced the *N*, *N*-diphenylpyrimidine capping group of ACY1215 with 5, 6, 7, 8-tetrahydropyrido[3, 4-*d*]pyrimidine and retained the six-carbon linker as well as hydroxamic acid ZBG (Figure 2). Here, we reported the design, structure, and activity relationship (SAR) study and antiproliferative evaluation of these tetrahydropyridopyrimidines.

## 2. Results and Discussion

### 2.1. Chemistry

The synthetic route to target compounds **8a**–**h** was initiated by the preparation of key intermediate **6a**–**h** from commercially available material **5** and two equivalent arylboronic acids by Suzuki reaction with Pd(dppf)Cl_2_ as a catalyst and K_2_CO_3_ as a base (Figure 1). For preliminary exploration, the same aryls were introduced on the C2 and C4 positions of the tetrahydropyridopyrimidine scaffold. Compounds **6a**–**h** underwent Boc deprotection under TFA/CH_2_Cl_2_ condition and subsequent condensation reaction with 8-methoxy-8-oxooctanoic acid through HATU, yielding the ester precursors **7a**–**h**. Then, **7a**–**h** was converted to the final hydroxamate product **8a**–**h** using aqueous hydroxylamine under basic conditions. Different electron-withdrawing or electron-donating substituents were introduced on two phenyls present in the capping part to explore the SAR. Moreover, the phenyl group was also replaced with an aromatic heterocycle such as thienyl or furyl.

### 2.2. HDAC1, 6 Activities and SAR Study of the Target Compounds

The target compounds **8a**–**h** were screened against HDAC6 with sHDAC6i ACY1215 and nonselective SAHA as the positive controls. Considering specific and redundant functions of class I HDACs in the control of proliferation as well as potential toxicity [38], HDAC1 was chosen for selectivity evaluation. As displayed in Table 1, all eight compounds demonstrated low nanomolar HDAC6 activity and two-digital selectivity against HDAC1. The most potent **8f**, with *meta*-OMe phenyls as the capping group, inhibited HDAC6 with an IC_50_ value of 6.4 nM and showed 48-fold selectivity against HDAC1, better than that of ACY1215. In addition, unsubstituted **8a** also had an IC_50_ of 16.2 nM and 35-fold selectivity. The introduction of *para*-OMe phenyl (**8c**) maintained the potency. Although -CF_3_, -Me, or furyl were adopted, a slight decrease in HDAC6 inhibition was observed (**8b**, **8d**, and **8h**). For thienyl derivatives **8e** and **8g**, the position of the sulfur atom obviously affected the HDAC6 activity (25.7 nM vs. 54 nM, respectively). It seemed that the substituent on the phenyl cap was critical for enzymatic activity.

**8f**, with the highest potency, was chosen for a detailed screening against other HDACs, including class I HDACs (HDAC2, 3, 8), HDAC 4, 5 (class IIa), and HDAC6 (class IIb) with ACY1215, SAHA, and TMP269 (a selective class IIa inhibitor) [39] as references. As demonstrated in Table 2, **8f** shows highly selective inhibition (more than 48-fold over other subtypes) toward HDAC6, and its selectivity values were higher than those of reference compound ACY1215. The IC_50_ values of **8f** against HDAC1-3 were 308 nM, 390 nM, and 411 nM, respectively. **8f** showed poor activity for HDAC4, 5 and 8. The results further validate the importance of tetrahydropyridopyrimidine with bulky capping groups to yield pronounced HDAC6 selective inhibition.

### 2.3. Western Blot Assay

To further determine the intracellular target specificity of **8f**, human MM cell line RPMI-8226 was treated at concentrations of 1, 5, and 10 μM, along with the reference HDAC6i ACY1215 and pan-inhibitor SAHA at 10 μM (Figure 3). **8f** was able to increase the levels of acetylated *α*-tubulin in a dose-dependent manner while inducing only modest changes in the levels of acetylated histone 3 (H3), similar to those found for the reference HDAC6i ACY-1215 at 10 μM. As expected, the pan-active HDACi SAHA increased levels of both acetylated α-tubulin and acetylated histone H3 significantly compared to the vehicle.

### 2.4. Molecular Simulation

The representative **8a** was docked into the human HDAC6 protein complex to elucidate the interaction model between these tetrahydropyridopyrimidines and the target protein. As outlined in Figure 4A, hydroxamate-Zn^2+^ coordination was modeled with bidentate geometry, and the Zn^2+^−O distances are 2.4 and 1.8 Å for the OH and C=O groups, respectively. The side chain of His610 additionally accepted a hydrogen bond from the hydroxamate OH group. The aliphatic chain linker embeds into the channel between Phe620 and Phe680. Moreover, two phenyl substituents of **8a** in the cap region were oriented into the crevice formed by Met682, Asp567, and Ser564. Tetrahydropyridopyrimidine scaffold as a proper connecting unit made the capping group of **8a** match well with amino acids on the rim of the binding tunnel (Figure 4B). For compound **8f**, its polar *meta*-OMe group improved the HDAC6 activity. For comparison, ACY-1215 was also docked into the HDAC6 crystallographic structure, and the superimposition of ACY-1215 and **8a** was disclosed in Figure 4C. Both compounds occupied the same pocket and showed similar binding modes.

### 2.5. Antiproliferative Activities of Representative Compounds

Hematological tumors such as lymphoma, multiple myeloma, and chronic myeloid leukemia are more sensitive to HDAC inhibitors. Therefore, HL60 and RPMI-8226 tumor cells were used for antiproliferative biological tests of our compounds. Moreover, colon cancer cell HCT116 was also added to evaluate the antiproliferative effect for solid tumors of these tetrahydropyridopyrimidines. IC_50_ values of three representative compounds **8a**, **8c**, and **8f** toward HL60 and RPMI-8226 cells range from 2.80 to 16.3 μM, which indicated that these tetrahydropyridopyrimidines tested kept the cell-based activity (Table 3). For solid tumor cells HCT116, all three analogs exhibited promising efficacy, especially for **8c** and **8f**, with IC_50_s of 4.72 and 3.25 μM. This result rendered these new inhibitors valuable hits for applications beyond multiple myeloma.

Then, **8f** was submitted to NCI for antiproliferative evaluation against 59 different tumor cell lines. The cancer types of the NCI-60 program include leukemia, non-small cell lung cancer (NSCLC), colon cancer, CNS cancer, melanoma, ovary cancer, renal cancer, prostate cancer, and breast cancer. As shown in Table 4, **8f** had an overall antiproliferative profile with percent inhibitions of 56 cell lines > 80% at 10 μM concentration.

## 3. Experimental Section

### 3.1. Chemistry

All the starting reagents were purchased and were used with no additional purification. All the mentioned yields were for isolated products. Melting points were determined in open capillaries on a WRS-1A digital melting point apparatus (Shenguang). ^1^H-NMR spectra were detected on a Bruker DRX–400 (400 MHz) using TMS as the internal standard. High-resolution mass spectra were obtained from Thermo Scientific Q Exactive. The chemical shifts were reported in ppm (*δ*), and coupling constants (*J*) values were given in Hertz (Hz). The purities of all target compounds were tested by HPLC to be >95.0%. HPLC analysis was performed at room temperature using an Agilent Eclipse XDB-C18 (250 mm × 4.6 mm) and plotted at 254 nm by 30% MeOH/H_2_O as a mobile phase.

#### 3.1.1. Tert-Butyl 2, 4-Diphenyl-5, 8-dihydropyrido[3, 4-*d*]pyrimidine-7(6*H*)-carboxylate (**6a**)

To a stirred mixture of tert-butyl 2, 4-dichloro-5, 8-dihydropyrido[3, 4-*d*]pyrimidine-7(6*H*)-carboxylate (304.2 mg, 1 mmol), potassium carbonate (345.5 mg, 2.5 mmol) and Pd(dppf)Cl_2_ (36.3 mg, 0.05 mmol) in 50 mL of 1, 4-dioxane was added phenylboronic acid (243.9 mg, 2 mmol). After stirring at reflux for 8 h under an argon atmosphere, the reaction mixture was concentrated under reduced pressure. Then, the reaction mixture was diluted with saturated sodium chloride (100 mL) and extracted with EtOAc (100 mL × 3). The combined organic extracts were dried over anhydrous Na_2_SO_4_ and concentrated under reduced pressure. The white product was obtained by chromatography on a silica gel column with a yield of 96%. ^1^H-NMR (400 MHz, DMSO-*d*_6_) *δ*: 8.44–8.38 (m, 2H), 7.81–7.75 (m, 2H), 7.57–7.50 (m, 6H), 4.67 (s, 2H), 3.57 (s, 2H), 2.88 (t, *J* = 5.4 Hz, 2H), 1.47 (s, 9H).

#### 3.1.2. Tert-Butyl 2, 4-di-p-tolyl-5, 8-dihydropyrido[3, 4-*d*]pyrimidine-7(6*H*)-carboxylate (**6b**)

**6b** was synthesized from (4-methylphenyl)boronic acid using a procedure similar to that described for the synthesis of **6a** and was obtained as a white solid (yield: 88%). ^1^H-NMR (400 MHz, DMSO-*d*_6_) *δ*: 8.32 (d, *J* = 8.9 Hz, 2H), 7.75 (d, *J* = 8.8 Hz, 2H), 7.07 (dd, *J* = 12.9, 8.9 Hz, 4H), 4.61 (s, 2H), 2.42 (s, 3H), 2.35 (s, 3H), 3.54 (s, 2H), 2.86 (t, *J* = 5.4 Hz, 2H), 1.47 (s, 9H).

#### 3.1.3. Tert-Butyl 2, 4-Bis(4-methoxyphenyl)-5, 8-dihydropyrido[3, 4-*d*]pyrimidine-7(6*H*)-carboxylate (**6c**)

**6c** was synthesized from (4-methoxyphenyl)boronic acid using a procedure similar to that described for the synthesis of **6a** and was obtained as a white solid (yield: 90%).^1^H-NMR (400 MHz, DMSO-*d*_6_) *δ*: 8.35 (d, *J* = 8.9 Hz, 2H), 7.77 (d, *J* = 8.8 Hz, 2H), 7.07 (dd, *J* = 12.9, 8.9 Hz, 4H), 4.62 (s, 2H), 3.84 (d, *J* = 6.1 Hz, 6H), 3.56 (s, 2H), 2.88 (t, *J* = 5.4 Hz, 2H), 1.47 (s, 9H).

#### 3.1.4. Tert-Butyl 2, 4-Di(furan-3-yl)-5, 8-dihydropyrido[3, 4-*d*]pyrimidine-7(6*H*)-carboxylate (**6d**)

Similar to the synthesis of **6a**, **6d** was obtained from furan-3-ylboronic acid as a white solid (yield: 88%). ^1^H-NMR (400 MHz, DMSO-*d*_6_) *δ*: 8.43 (d, *J* = 12.1 Hz, 2H), 7.86 (s, 1H), 7.79 (s, 1H), 7.23 (s, 1H), 7.08 (s, 1H), 4.56 (s, 2H), 3.65 (s, 2H), 2.88 (t, *J* = 5.4 Hz, 2H), 1.45 (s, 9H).

#### 3.1.5. Tert-Butyl 2, 4-Di(thiophen-3-yl)-5,8-dihydropyrido [3, 4-*d*]pyrimidine-7(6*H*)-carboxylate (**6e**)

Similar to the synthesis of **6a**, **6e** was obtained from thiophen-3-ylboronic acid as a white solid (yield: 87%). ^1^H-NMR (400 MHz, DMSO-*d*_6_) *δ*: 8.37 (d, *J* = 3.9 Hz, 1H), 8.24–8.20 (m, 1H), 7.83 (d, *J* = 6.0 Hz, 1H), 7.75 (d, *J* = 5.0 Hz, 1H), 7.71 (dd, *J* = 5.0, 2.8 Hz, 1H), 7.64 (dd, *J* = 5.0, 3.1 Hz, 1H), 4.60 (s, 2H), 3.61 (s, 2H), 2.97 (t, *J* = 5.4 Hz, 2H), 1.46 (s, 9H).

#### 3.1.6. Tert-Butyl 2, 4-Bis(3-methoxyphenyl)-5, 8-dihydropyrido [3, 4-*d*]pyrimidine-7(6*H*)-carboxylate (**6f**)

Similar to the synthesis of **6a**, **6f** was obtained from (3-methoxyphenyl)boronic acid as a white solid (yield: 89%). ^1^H-NMR (400 MHz, DMSO-*d*_6_) *δ*: 8.00 (d, *J* = 7.8 Hz, 1H), 7.92 (s, 1H), 7.46-7.44 (m, 2H), 7.33–7.26 (m, 2H), 7.12-7.10 (m, 2H), 4.67 (s, 2H), 3.83 (s, 6H), 3.57 (s, 2H), 2.87 (t, *J* = 5.4 Hz, 2H), 1.47 (s, 9H).

#### 3.1.7. Tert-Butyl 2, 4-Di(thiophen-2-yl)-5, 8-dihydropyrido [3, 4-*d*]pyrimidine-7(6*H*)-carboxylate (**6g**)

Similar to the synthesis of **6a**, **6g** was obtained from thiophen-2-ylboronic acid as a white solid (yield: 85%). ^1^H-NMR (400 MHz, DMSO-*d*_6_) *δ*: 7.94 (dd, *J* = 3.6, 1.1 Hz, 1H), 7.89 (d, *J* = 5.1 Hz, 1H), 7.85 (d, *J* = 3.7 Hz, 1H), 7.76 (dd, *J* = 5.0, 1.0 Hz, 1H), 7.30–7.27 (m, 1H), 7.23–7.20 (m, 1H), 4.59 (s, 2H), 3.67 (s, 2H), 3.04 (t, *J* = 5.5 Hz, 2H), 1.46 (s, 9H).

#### 3.1.8. Tert-Butyl 2, 4-Bis(4-(trifluoromethyl)phenyl)-5, 8-dihydropyrido [3, 4-*d*]pyrimidine-7(6*H*)-carboxylate (**6h**)

Similar to the synthesis of **6a**, **6h** was obtained from (4-(trifluoromethyl)phenyl)boronic acid as a white solid (yield: 92%). ^1^H-NMR (400 MHz, DMSO-*d*_6_) *δ*: 8.60 (d, *J* = 8.2 Hz, 2H), 8.01 (d, *J* = 8.2 Hz, 2H), 7.95–7.84 (m, 4H), 4.72 (s, 2H), 3.59 (s, 2H), 2.90 (t, *J* = 5.4 Hz, 2H), 1.47 (s, 9H).

#### 3.1.9. Methyl 8-(2, 4-Diphenyl-5, 8-dihydropyrido [3, 4-*d*]pyrimidin-7(6*H*)-yl)-8-oxooctanoate (**7a**)

(i) To a stirred mixture of **6a** (387.5 mg, 1 mmol) in CH_2_Cl_2_ (40 mL) was added TFA (5 mL) in portions. The reaction mixture was stirred at reflux for 2 h and then concentrated under reduced pressure. The reaction mixture was diluted with saturated sodium chloride (50 mL) and adjusted to PH = 7 with Na_2_CO_3_ saturated solution. Then, the mixture was extracted with EtOAc (50 mL × 3). The combined organic extracts were dried over anhydrous Na_2_SO_4_ and concentrated under reduced pressure. The product was obtained as an oil. (ii) To a solution of the product acquired in step (i) in DMF (30 mL) was added HATU (1 mmol) DIPEA (4 mmol) at 0 °C. Then, the reaction mixture was stirred at room temperature overnight. After the completion of the reaction detected by TLC, the reaction was poured into water (30 mL) and extracted with EtOAc (60 mL × 3). The combined organic extracts were dried over anhydrous Na_2_SO_4_ and concentrated under reduced pressure. Then, the resulting mixture was purified by column chromatography to give the product **7a** with 54% isolated yield: white solid. ^1^H-NMR (400 MHz, DMSO-*d*_6_) *δ*: 8.41 (s, 2H), 7.77 (d, *J* = 3.5 Hz, 2H), 7.59–7.46 (m, 6H), 4.79 (d, *J* = 21.8 Hz, 2H), 3.67 (t, *J* = 5.1 Hz, 2H), 3.57 (s, 3H), 2.93 (t, *J* = 5.2 Hz, 1H), 2.82 (t, *J* = 5.2 Hz, 1H), 2.49–2.37 (m, 2H), 2.32–2.22 (m, 2H), 1.53 (m, 4H), 1.30 (d, *J* = 3.6 Hz, 4H).

#### 3.1.10. Methyl 8-(2, 4-di-p-tolyl-5, 8-dihydropyrido [3, 4-*d*]pyrimidin-7(6*H*)-yl)-8-oxooctanoate (**7b**)

Similar to the synthesis of **7a**, **7b** was obtained as a white solid with a yield of 49%. ^1^H-NMR (400 MHz, DMSO-*d*_6_) *δ*: 8.31 (dd, *J* = 8.2, 2.0 Hz, 2H), 7.68 (dd, *J* = 8.1, 2.4 Hz, 2H), 7.34 (dd, *J* = 14.1, 8.1 Hz, 4H), 4.78 (d, *J* = 28.1 Hz, 2H), 3.68 (t, *J* = 5.4 Hz, 2H), 3.57 (s, 3H), 2.94 (t, *J* = 5.2 Hz, 1H), 2.83 (t, *J* = 5.2 Hz, 1H), 2.48–2.42 (m, 2H), 2.40 (s, 3H), 2.38 (s, 3H), 2.31-2.28 (m, 2H), 1.58–1.47 (m, 4H), 1.35–1.22 (m, 4H).

#### 3.1.11. Methyl 8-(2, 4-Bis(4-methoxyphenyl)-5, 8-dihydropyrido [3, 4-*d*]pyrimidin-7(6*H*)-yl)-8-oxooctanoate (**7c**)

Similar to the synthesis of **7a**, **7c** was obtained as a yellow solid with a yield of 38%. ^1^H-NMR (400 MHz, DMSO-*d*_6_) *δ*: 8.35 (d, *J* = 8.5 Hz, 2H), 7.76 (d, *J* = 8.0 Hz, 2H), 7.07 (dd, *J* = 12.9, 8.5 Hz, 4H), 4.74 (d, *J* = 26.0 Hz, 2H), 3.83 (m, 6H), 3.67 (s, 2H), 3.57 (s, 3H), 2.94 (s, 1H), 2.83 (s, 1H), 2.43 (dd, *J* = 12.2, 6.4 Hz, 2H), 2.29 (t, *J* = 5.7 Hz, 2H), 1.53–1.51 (m, 4H), 1.34–1.25 (m, 4H).

#### 3.1.12. Methyl 8-(2, 4-Di(furan-3-yl)-5, 8-dihydropyrido [3, 4-*d*]pyrimidin-7(6*H*)-yl)-8-oxooctanoate (**7d**)

Similar to the synthesis of **7a**, **7d** was obtained as a yellow solid with a yield of 43%. ^1^H-NMR (400 MHz, DMSO-*d*_6_) *δ*: 8.44 (s, 1H), 8.23–8.10 (m, 2H), 7.85 (s, 1H), 7.67 (d, *J* = 4.2 Hz, 1H), 7.20 (d, *J* = 3.0 Hz, 1H), 7.05 (s, 1H), 4.66 (d, *J* = 16.7 Hz, 2H), 3.77 (s, 2H), 3.56 (s, 3H), 2.94 (s, 1H), 2.83 (s, 1H), 2.42 (m, 2H), 1.91 (q, *J* = 6.9 Hz, 2H), 1.45 (dd, *J* = 16.1, 9.0 Hz, 4H), 1.38–1.15 (m, 4H).

#### 3.1.13. Methyl 8-(2, 4-Di(thiophen-3-yl)-5, 8-dihydropyrido [3, 4-*d*]pyrimidin-7(6*H*)-yl)-8-oxooctanoate (**7e**)

Similar to the synthesis of **7a**, **7e** was obtained as a white solid with a yield of 41%. ^1^H-NMR (400 MHz, DMSO-*d*_6_) *δ*: 8.38 (dd, *J* = 3.0, 1.1 Hz, 1H), 8.23–8.18 (m, 1H), 7.84 (d, *J* = 5.0 Hz, 1H), 7.78–7.69 (m, 2H), 7.66–7.62 (m, 1H), 4.73 (d, *J* = 21.2 Hz, 2H), 3.73 (t, *J* = 4.3 Hz, 2H), 3.57 (d, *J* = 2.7 Hz, 3H), 3.04 (t, *J* = 5.3 Hz, 1H), 2.94 (t, *J* = 5.2 Hz, 1H), 2.44 (t, *J* = 7.4 Hz, 2H), 2.28 (q, *J* = 7.1 Hz, 2H), 1.52–1.50 (m, 4H), 1.34–1.24 (m, 4H).

#### 3.1.14. Methyl 8-(2, 4-Bis(3-methoxyphenyl)-5, 8-dihydropyrido [3, 4-*d*]pyrimidin-7(6*H*)-yl)-8-oxooctanoate (**7f**)

Similar to the synthesis of **7a**, **7f** was obtained as a yellow solid with a yield of 48%. ^1^H-NMR (400 MHz, DMSO-*d*_6_) *δ*: 8.46 (s, 1H), 7.98 (d, *J* = 7.8 Hz, 1H), 7.90 (s, 1H), 7.47–7.40 (m, 2H), 7.32–7.21 (m, 2H), 7.04 (m, 2H), 4.80 (d, *J* = 26.2 Hz, 2H), 3.82 (s, 6H), 3.65 (t, *J* = 5.5 Hz, 2H), 3.57 (s, 3H), 2.92 (t, *J* = 5.1 Hz, 1H), 2.82 (t, *J* = 5.1 Hz, 1H), 2.45–2.35 (m, 2H), 1.92 (t, *J* = 7.3 Hz, 2H), 1.50–1.48 (m, 4H), 1.26 (q, *J* = 16.1, 12.5 Hz, 4H).

#### 3.1.15. Methyl 8-(2, 4-Di(thiophen-2-yl)-5, 8-dihydropyrido [3, 4-*d*]pyrimidin-7(6*H*)-yl)-8-oxooctanoate (**7g**)

Similar to the synthesis of **7a**, **7g** was obtained as a yellow solid with a yield of 49%. ^1^H-NMR (400 MHz, DMSO-*d*_6_) *δ*: 8.36 (s, 1H), 7.90 (d, *J* = 4.5 Hz, 1H), 7.65 (d, *J* = 5.0 Hz, 1H), 7.77 (t, *J* = 4.3 Hz, 1H), 7.70 (d, *J* = 4.8 Hz, 1H), 7.26 (q, *J* = 5.0 Hz, 1H), 7.23–7.17 (m, 1H), 4.71 (d, *J* = 23.0 Hz, 2H), 3.83–3.70 (m, 2H), 3.57 (s, 3H), 3.11 (t, *J* = 5.1 Hz, 1H), 2.96 (t, *J* = 5.2 Hz, 1H), 2.47–2.42 (m, 2H), 1.93–1.91 (m, 2H), 1.56–1.41 (m, 4H), 1.33–1.16 (m, 4H).

#### 3.1.16. Methyl 8-(2, 4-Bis(4-(trifluoromethyl)phenyl)-5, 8-dihydropyrido [3, 4-*d*]pyrimidin-7(6*H*)-yl)-8-oxooctanoate (**7h**)

Similar to the synthesis of **7a**, **7g** was obtained as a yellow solid with a yield of 46%. ^1^H-NMR (400 MHz, DMSO-*d*_6_) *δ*: 8.34 (s, 1H), 8.19 (d, *J* = 8.2 Hz, 2H), 8.00 (d, *J* = 8.1 Hz, 2H), 7.90 (dd, *J* = 12.7, 8.3 Hz, 4H), 4.82 (d, *J* = 27.8 Hz, 2H), 3.71–3.60 (m, 2H), 2.92 (t, *J* = 4.8 Hz, 1H), 2.81 (t, *J* = 4.7 Hz, 1H), 2.42–2.33 (m, 2H), 1.93 (t, *J* = 7.3 Hz, 2H), 1.51–1.49 (m, 4H), 1.36–1.15 (m, 4H).

#### 3.1.17. 8-(2, 4-Diphenyl-5, 8-dihydropyrido [3, 4-*d*]pyrimidin-7(6*H*)-yl)-*N*-hydroxy-8-oxooctanamide (**8a**)

A solution of NH_2_OH·HCl (1.70 g, 24.46 mmol) in MeOH (9 mL) was combined with KOH (1.70 g, 30.29 mmol) at 0 °C in an ice bath. Then, the mixture was stirred for 30 min and filtered. **7a** (457.6 mg, 1 mmol) was added to the filtrate, and the reaction was stirred for an additional 4 h at 0 °C in an ice bath. The resulting mixture was poured into water (30 mL), and the pH value was adjusted to 7. The mixture was diluted with saturated NaCl aqueous solution (40 mL) and extracted with EtOAc (50 mL × 3). After drying over Na_2_SO_4_, the organic phase was concentrated and purified by column chromatography to give the product **8a**. 84% yield; white solid; m.p.: 115~117 °C. ^1^H-NMR (400 MHz, DMSO-*d*_6_) *δ*: 10.35 (s, 1H), 8.67 (s, 1H), 8.45-8.39 (m, 2H), 7.80-7.74 (m, 2H), 7.56-7.52 (m, 6H), 4.80 (d, *J* = 24.4 Hz, 2H), 3.69 (t, *J* = 5.5 Hz, 2H), 2.95 (t, *J* = 5.1 Hz, 1H), 2.84 (t, *J* = 5.1 Hz, 1H), 2.48–2.40 (m, 2H), 1.95 (t, *J* = 7.3 Hz, 2H), 1.53–1.51 (m, 4H), 1.34–1.22 (m, 4H). ^13^C-NMR (101 MHz, DMSO-*d*_6_) *δ*: 171.24, 169.14, 164.54, 163.38, 160.67, 137.33, 137.03, 130.63, 129.61, 129.07, 128.64, 128.36, 127.62, 123.75, 46.44, 42.13, 32.66, 32.26, 28.49, 26.70, 25.88, 25.06, 24.54. HR-MS (ESI, *m*/*z*): Calcd for 459.23907 (C_27_H_31_N_4_O_3_^+^ [M + H]^+^). Found 459.23901.

#### 3.1.18. 8-(2, 4-di-p-tolyl-5, 8-dihydropyrido [3, 4-*d*]pyrimidin-7(6*H*)-yl)-N-hydroxy-8-oxooctanamide (**8b**)

Similar to the synthesis of **8a**, **8b** was obtained from **7b** as a white solid (yield: 64%). M.p.: 110~112 °C. ^1^H-NMR (400 MHz, DMSO-*d*_6_) *δ*: 10.35 (s, 1H), 8.68 (s, 1H), 8.31 (dd, *J* = 8.2, 2.2 Hz, 2H), 7.69 (dd, *J* = 8.1, 2.7 Hz, 2H), 7.37 (s, 1H), 7.36–7.33 (m, 2H), 7.32 (d, *J* = 1.9 Hz, 1H), 4.78 (d, *J* = 28.7 Hz, 2H), 3.68 (t, *J* = 4.9 Hz, 2H), 2.95 (t, *J* = 5.2 Hz, 1H), 2.84 (t, *J* = 5.3 Hz, 1H), 2.45 (d, *J* = 7.1 Hz, 2H), 2.41 (s, 3H), 2.38 (s, 3H), 1.94 (t, *J* = 7.3 Hz, 2H), 1.51 (m, 4H), 1.33–1.24 (m, 4H). ^13^C-NMR (101 MHz, DMSO-*d*_6_) *δ*: 171.21, 169.16, 164.34, 163.14, 160.68, 140.35, 139.36, 139.32, 134.57, 134.44, 129.23, 129.07, 128.90, 127.59, 123.35, 123.22, 46.45, 42.19, 32.27, 28.50, 26.78, 25.08, 24.55, 24.44, 20.96. HR-MS (ESI, *m*/*z*): Calcd for 487.27037 (C_29_H_35_N_4_O_3_^+^ [M + H]^+^). Found 487.27020.

#### 3.1.19. 8-(2, 4-Bis(4-methoxyphenyl)-5, 8-dihydropyrido [3, 4-*d*]pyrimidin-7(6*H*)-yl)-*N*-hydroxy-8-oxooctanamide (**8c**)

Similar to the synthesis of **8a**, **8c** was obtained from **7c** as a white solid (yield: 57%). M.p.: 117~118 °C. ^1^H-NMR (400 MHz, DMSO-*d*_6_) *δ*: 10.35 (s, 1H), 8.68 (s, 1H), 8.38 (d, *J* = 2.0 Hz, 1H), 8.35 (d, *J* = 2.0 Hz, 1H), 7.77 (d, *J* = 8.6 Hz, 2H), 7.12–7.10 (m, 1H), 7.09–7.06 (m, 2H), 7.05 (d, *J* = 2.6 Hz, 1H), 4.76 (d, *J* = 28.2 Hz, 2H), 3.85 (s, 3H), 3.83 (s, 3H), 3.68 (t, *J* = 5.3 Hz, 2H), 2.96 (t, *J* = 4.9 Hz, 1H), 2.85 (t, *J* = 5.1 Hz, 1H), 2.45 (dd, *J* = 13.6, 6.6 Hz, 2H), 1.94 (t, *J* = 7.3 Hz, 2H), 1.51 (m, 4H), 1.34–1.21 (m, 4H). ^13^C-NMR (101 MHz, DMSO-*d*_6_) *δ*: 169.12, 161.96, 161.34, 160.43, 144.74, 138.38, 137.87, 130.83, 130.78, 129.69, 129.65, 129.23, 120.97, 120.82, 113.96, 113.73, 46.46, 46.07, 32.25, 28.49, 28.46, 26.87, 25.05, 24.54, 24.42. HR-MS (ESI, *m*/*z*): Calcd for 519.26020 (C_29_H_35_N_4_O_5_^+^ [M + H]^+^). Found 519.26001.

#### 3.1.20. 8-(2, 4-Di(furan-3-yl)-5, 8-dihydropyrido [3, 4-*d*]pyrimidin-7(6*H*)-yl)-*N*-hydroxy-8-oxooctanamide (**8d**)

Similar to the synthesis of **8a**, **8d** was obtained from **7d** as a white solid (yield: 65%). M.p.: 104~106 °C. ^1^H-NMR (400 MHz, DMSO-*d*_6_) *δ*: 10.35 (s, 1H), 8.70 (s, 1H), 8.46–8.35 (m, 2H), 7.87 (s, 1H), 7.80 (d, *J* = 4.2 Hz, 1H), 7.23 (d, *J* = 3.0 Hz, 1H), 7.08 (s, 1H), 4.69 (d, *J* = 16.7 Hz, 2H), 3.77 (s, 2H), 2.96 (s, 1H), 2.85 (s, 1H), 2.43 (q, *J* = 7.0 Hz, 2H), 1.95–1.93 (m, 2H), 1.50 (dd, *J* = 16.1, 9.0 Hz, 4H), 1.39–1.17 (m, 4H). ^13^C-NMR (101 MHz, DMSO-*d*_6_) *δ*: 171.16, 171.09, 169.20, 166.19, 162.91, 145.19, 144.70, 144.46, 143.81, 126.54, 124.08, 122.12, 121.94, 46.25, 41.98, 32.58, 32.28, 32.05, 28.48, 26.38, 25.07, 24.57. HR-MS (ESI, *m*/*z*): Calcd for 439.19760 (C_29_H_35_N_4_O_5_^+^ [M + H]^+^). Found 439.19751.

#### 3.1.21. 8-(2, 4-Di(thiophen-3-yl)-5,8-dihydropyrido [3, 4-*d*]pyrimidin-7(6*H*)-yl)-*N*-hydroxy-8-oxooctanamide (**8e**)

Similar to the synthesis of **8a**, **8e** was obtained from **7e** as a white solid (yield: 68%). M.p.: 123~124 °C. ^1^H-NMR (400 MHz, DMSO-*d*_6_) *δ*: 10.35 (s, 1H), 8.68 (s, 1H), 8.40–8.36 (m, 1H), 8.22 (d, *J* = 8.8 Hz, 1H), 7.84 (d, *J* = 5.0 Hz, 1H), 7.78–7.69 (m, 2H), 7.65 (m, 1H), 4.74 (d, *J* = 21.9 Hz, 2H), 3.73 (t, *J* = 5.4 Hz, 2H), 3.05 (t, *J* = 5.0 Hz, 1H), 2.94 (t, *J* = 5.0 Hz, 1H), 2.45 (t, *J* = 7.2 Hz, 2H), 2.00–1.87 (m, 2H), 1.61–1.40 (m, 4H), 1.36–1.21 (m, 4H). ^13^C-NMR (101 MHz, DMSO-*d*_6_) *δ*: 171.18, 171.09, 169.16, 163.39, 163.09, 141.12, 138.75, 129.23, 129.00, 128.77, 127.87, 127.17, 127.01, 126.35, 46.42, 42.14, 38.03, 32.27, 32.11, 26.79, 26.02, 25.06, 24.55. HR-MS (ESI, *m*/*z*): Calcd for 471.15191 (C_23_H_27_N_4_O_3_S_2_^+^ [M + H]^+^). Found 471.15195.

#### 3.1.22. 8-(2, 4-Bis(3-methoxyphenyl)-5, 8-dihydropyrido [3, 4-*d*]pyrimidin-7(6*H*)-yl)-*N*-hydroxy-8-oxooctanamide (**8f**)

Similar to the synthesis of **8a**, **8f** was obtained from **7f** as a white solid (yield: 64%). M.p.: 117~118 °C. ^1^H-NMR (400 MHz, DMSO-*d*_6_) *δ*: 10.34 (s, 1H), 8.67 (s, 1H), 8.00 (d, *J* = 7.8 Hz, 1H), 7.93 (s, 1H), 7.50–7.40 (m, 2H), 7.33–7.25 (m, 2H), 7.10 (td, *J* = 8.1, 2.2 Hz, 2H), 4.80 (d, *J* = 26.2 Hz, 2H), 3.83 (s, 6H), 3.69 (t, *J* = 5.5 Hz, 2H), 2.94 (t, *J* = 5.1 Hz, 1H), 2.83 (t, *J* = 5.1 Hz, 1H), 2.48–2.39 (m, 2H), 1.94 (t, *J* = 7.3 Hz, 2H), 1.52–1.50 (m, 4H), 1.27–1.25 (m, 4H). ^13^C-NMR (101 MHz, DMSO-*d*_6_) *δ*: 171.25, 169.14, 164.41, 163.35, 160.40, 159.57, 159.15, 138.68, 138.50, 129.78, 129.55, 123.96, 121.26, 120.09, 116.35, 115.34, 115.13, 114.54, 114.39, 112.67, 46.44, 42.10, 32.26, 32.14, 28.49, 26.68, 25.88, 25.07, 24.54. HR-MS (ESI, *m*/*z*): Calcd for 519.26020 (C_29_H_35_N_4_O_5_^+^ [M + H]^+^). Found 519.26019.

#### 3.1.23. 8-(2, 4-Di(thiophen-2-yl)-5, 8-dihydropyrido [3, 4-*d*]pyrimidin-7(6*H*)-yl)-*N*-hydroxy-8-oxooctanamide (**8g**)

Similar to the synthesis of **8a**, **8g** was obtained from **7g** as a white solid (yield: 71%). M.p.: 128~129 °C. ^1^H-NMR (400 MHz, DMSO-*d*_6_) *δ*: 10.34 (s, 1H), 8.66 (s, 1H), 7.94 (d, *J* = 4.5 Hz, 1H), 7.90 (d, *J* = 5.0 Hz, 1H), 7.83 (t, *J* = 4.3 Hz, 1H), 7.76 (d, *J* = 4.8 Hz, 1H), 7.29 (q, *J* = 5.0 Hz, 1H), 7.24–7.19 (m, 1H), 4.73 (d, *J* = 23.0 Hz, 2H), 3.86–3.72 (m, 2H), 3.12 (t, *J* = 5.1 Hz, 1H), 2.99 (t, *J* = 5.2 Hz, 1H), 2.48–2.43 (m, 2H), 1.95–1.93 (m, 2H), 1.57–1.43 (m, 4H), 1.34–1.20 (m, 4H). ^13^C-NMR (101 MHz, DMSO-*d*_6_) *δ*: 171.15, 169.12, 163.90, 163.60, 157.29, 142.32, 141.57, 131.36, 131.12, 130.96, 130.56, 128.86, 128.59, 128.50, 49.31, 46.34, 32.25, 32.02, 28.45, 26.84, 26.03, 25.05, 24.54. HR-MS (ESI, *m*/*z*): Calcd for 471.15191 (C_23_H_27_N_4_O_3_S_2_^+^[M + H]^+^). Found 471.15192.

#### 3.1.24. 8-(2, 4-Bis(4-(trifluoromethyl)phenyl)-5, 8-dihydropyrido [3, 4-*d*]pyrimidin-7(6*H*)-yl)-*N*-hydroxy-8-oxooctanamide (**8h**)

Similar to the synthesis of **8a**, **8h** was obtained from **7h** as a white solid (yield: 64%). M.p.: 145~146 °C. ^1^H-NMR (400 MHz, DMSO-*d*_6_) *δ*: 10.33 (s, 1H), 8.65 (s, 1H), 8.61 (d, *J* = 8.2 Hz, 2H), 8.02 (d, *J* = 8.1 Hz, 2H), 7.92 (dd, *J* = 12.7, 8.3 Hz, 4H), 4.87 (d, *J* = 27.8 Hz, 2H), 3.77–3.66 (m, 2H), 2.97 (t, *J* = 4.8 Hz, 1H), 2.87 (t, *J* = 4.7 Hz, 1H), 2.48–2.38 (m, 2H), 1.94 (t, *J* = 7.3 Hz, 2H), 1.52 (m, 4H), 1.37–1.17 (m, 4H). ^13^C-NMR (101 MHz, DMSO-*d*_6_) *δ*: 169.12, 164.17, 163.60, 163.52, 159.39, 142.42, 130.07, 130.05, 130.02, 129.98, 128.36, 125.73, 125.69, 125.37, 125.34, 46.39, 41.93, 32.24, 32.22, 32.09, 28.48, 28.44, 25.04, 24.50. HR-MS (ESI, *m*/*z*): Calcd for 595.21384 (C_29_H_29_F_6_N_4_O_3_^+^[M + H]^+^). Found 595.21344.

### 3.2. In Vitro HDAC Enzyme Assay

IC_50_ testing of compounds was performed by the Reaction Biology Corporation. The procedure was conducted as described previously [40].

### 3.3. Cell Culture and Antiproliferative Assay

The cells were cultured in IMDM (Gibco) medium with 20% FBS (Lonsera), 100 U/mL penicillin, and 100 µg/mL streptomycin (Solarbio). All cells were maintained at 37 °C in a humidified atmosphere of 5% CO_2_ in air. Briefly, 100 µL cell suspension or completed medium was plated into a 96-well plate. Compounds were added and incubated for 72 h. Then, 22 µL Alamar blue solution (1 mM) was pipetted into each well of a 96-well plate, and the plate was incubated for an additional 5~6 h. The absorbance (OD) was read at 530/590 nm. Data were normalized to vehicle groups (DMSO) and represented as the means of three independent measurements with standard errors of <20%. The IC_50_ values were calculated using Prism 5.0.

### 3.4. Western Blotting Assay

RPMI-8226 cells (1 × 10^6^) were seeded overnight and incubated with compound **8f** for 24 h on indicated concentrations. Cell extract was prepared by lysing cultured cells with a mammalian protein extraction reagent supplemented with EDTA-free protease inhibitor for 15 min. SDS-PAGE and immunoblot analysis were conducted as described [40]. Antibodies for Ac-H3 (abcom, AB32129) and Ac-*ɑ*-tubulin (Cell Signaling, 2144) were used.

### 3.5. Computational Methods

Molecular simulation was performed in Discovery Studio 3.0 software (BIOVIA, 5005 Wateridge Vista Drive, San Diego, CA, USA). Docking was conducted using cdocker based on the cocrystal of HDAC6 (PDB: 5EDU). The cavity occupied by trichostatin A was selected as the ligand binding site. The parameter setting was performed as previously reported [2].

## 4. Conclusions

In this work, a series of novel hybrid HDAC6 inhibitors were designed by chemically merging the structure of tetrahydropyridopyrimidine into the pharmacophore of HDAC6is. All newly synthesized compounds were first evaluated for inhibition of HDAC1 and HDAC6. All these diarylpyrimidine derivatives demonstrated potent HDAC6 activity at a nanomolar level and 16~49-fold selectivity over HDAC1. Western blot study further confirmed HDAC6 selectivity of these tetrahydropyridopyrimidines. In the cytotoxic assay, compounds **8a**, **8c**, and **8f** showed potent antiproliferative activity against representative hematological and solid tumors. Taken together, this work highlights the application of tetrahydropyridopyrimidine scaffold in the development of novel sHDAC6 inhibitors. These tetrahydropyridopyrimidine derivatives might be developed as new antitumor agents besides multiple myeloma. Further structural modification was performed in our lab.

## Data Availability

Data are contained within the article and Appendix A.

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
