# Peer review of "Design, Synthesis, and Antiproliferative Activity of Selective Histone Deacetylases 6 Inhibitors Containing a Tetrahydropyridopyrimidine Scaffold"

_molecules, 2023, doi:10.3390/molecules28217323_

Round 1

Reviewer 1 Report

Comments and Suggestions for Authors

I would like to express my appreciation to Bin et al. for their contribution to the development of histone deacetylase 6 inhibitors. The synthetic work was tremendous and well-described. The result of compound 8f showing excellent inhibition property is exciting. While the majority of the results are consistent and supportive of the proposed approach, I have the following concerns:

(1) Looks like the synthetic sHDAC6is is a new topic.  Only three papers (two are from the same author) are cited for this. Could you expand more on the rationale of the importance and difficulty of this field?

(2) Could you kindly provide the 13C NMR for all the precursors if they're available? The characterization data are sometimes pretty valuable for other researchers who want to take advantage of your work. Please also consider attaching their spectrum in the ESI. 

(3) 13C spectra of compounds 8c and 8h have a high S/N ratio due to the low concentration. Please try with higher concentration and more scans.

(4) The way of presenting R groups in Table 1 is confusing. Please either show only structures of R or change the table contents.

Reviewer 2 Report

Comments and Suggestions for Authors

This manuscript describes the development of HDAC6 inhibitors with Tetrahydropyridopyrimidine Scaffold. The authors focused on a big capping group of ACY-1215,  and designed compounds 8a-h which have tetrahydropyridopyrimidine instead of N, N-diphenylpyrimidine capping group. Among a series of tetrahydropyridopyrimidine-type compounds, 8f indicated the most potent HDAC6 inhibitory activity and superior selectivity index (HDAC1/HDAC6). Moreover, 8f showed a stronger antiproliferative effect than the original compound (ACY-1215) in HCT116. These results might be valuable for the development of HDAC inhibitors and anticancer agents. Thus this manuscript is recommended for publication on “Molecules” after minor revision.

Minor revision:

1.       Table 1: There are missing structures. Please make sure.

2.       Page 5, lines 142 144: Are there water molecules at the crevice formed by Met682, Asp567 and Ser564 in the original PDB data? If not, I think that the interaction between meta-OMe group and the polar solvent is not convincing for the reasons for the improvement of HDAC inhibitory activity. 
